# An Easy-to-Implement Risk Score for Targeted Hepatitis C Virus Testing in the General Population

Javier Martínez-Sanz,[a,b] María Jesús Vivancos-Gallego,[a,b] Borja Manuel Fernández-Felix,[c,d] Alfonso Muriel,[c,d,e] Pilar Pérez-Elías,[f] Almudena Uranga,[f] Beatriz Romero,[d,g] Juan Carlos Galán,[d,g] Santiago Moreno,[a,b,e] María Jesús Pérez-Elías[a,b]

[a]Department of Infectious Diseases, Hospital Universitario Ramón y Cajal, IRYCIS, Madrid, Spain
[b]CIBER de Enfermedades Infecciosas (CIBERINFEC), Madrid, Spain
[c]Clinical Biostatistics Unit, Hospital Universitario Ramón y Cajal, IRYCIS, Madrid, Spain
[d]CIBER de Epidemiología y Salud Pública (CIBERESP), Madrid, Spain
[e]Universidad de Alcalá, Madrid, Spain
[f]Centro de Salud García Noblejas, Madrid, Spain
[g]Department of Microbiology, Hospital Universitario Ramón y Cajal, IRYCIS, Madrid, Spain

**ABSTRACT** Despite the effectiveness of available treatments, hepatitis C virus (HCV) remains a major public health problem, mainly due to the high percentage of undiagnosed individuals. We aim to create an easy-to-implement risk score to facilitate targeted HCV testing in the general population. This is a substudy derived from a prospective study in primary care in Madrid (Spain). Participants completed a 21-question risk assessment questionnaire, followed by HCV testing for those with at least one positive response and those >50 years of age, even if they did not answer positively. We used the population >50 years of age to fit a logistic regression model to create a score predicting the risk of a positive test result. We then performed a sensitivity analysis by applying the score obtained to the population <50 years of age, to assess its diagnostic accuracy. Data collected from 2,302 participants were included in the analysis. The prevalence of HCV infection was 1.3%. Five items were selected, showing a C-statistic of 0.896, i.e., male sex, Eastern European origin, use of intravenous drugs, self-perceived risk of acquired HCV infection, and past hepatitis or unexplained liver disease. The sensitivity was 98%, and the negative likelihood ratio was 0.05 for participants with scores of 0 (49.8% in our sample), ruling out HCV infection with high probability. We obtained similar estimates in the population <50 years of age. This tool achieved high diagnostic accuracy to target HCV testing. This could help optimize resources when universal screening is not feasible.

**IMPORTANCE** Despite the highly effective treatments currently available, HCV remains one of the major public health problems related to an infectious agent, mainly because a high percentage of individuals remain undiagnosed. Universal screening has been proposed as a way to end this epidemic; however, it is not feasible in all settings due to different implementation barriers. With this work, we aim to collaborate in improving the diagnosis of HCV infection by creating a simple 5-item score that rules out HCV infection with a very high probability. Almost one-half of the participants in our sample did not present any affirmative answers to these questions, and their probability of being infected was close to 0%. This tool could be a useful strategy and could be considered a cost-effective alternative to optimize resources when universal screening is not feasible.

**KEYWORDS** hepatitis C virus, targeted screening, risk factors, score, testing

Address correspondence to Javier Martínez-Sanz, javier.martinez.sanz@salud.madrid.org.

The authors declare a conflict of interest. Javier Martínez-Sanz reports personal fees from ViiV Healthcare, Janssen Cilag, Gilead Sciences, and MSD and non-financial support from ViiV Healthcare, Jannsen Cilag, and Gilead Sciences, outside the submitted work. Beatriz Romero has received funding to attend conferences from Gilead and obtained a funded project from Roche. Juan Carlos Galán has been involved in speaking activities from Gilead Science, GeneExpert and Hologic. He received grants from Merck Sharp & Dohme, Abbvie, Abbott Molecular and Roche. He has done consulting works for Abbott Molecular and Gilead. Santiago Moreno has been involved in speaking activities and received grants for research from Abbott, Boehringer & Ingelheim, Bristol-Myers Squibb, Gilead, Glaxo Smith Kline, Janssen Cilag, Merck Sharp & Dohme, Pfizer, Roche, and Schering Plough. María Jesús Pérez-Elías has done consulting work for Abbvie, Boehringer Ingelheim, ViiV Healthcare, Gilead Sciences, and Janssen Cilag; she has received fellowships for clinical research from ViiV Healthcare, Gilead Sciences, Janssen, and Merck Sharp & Dome, and financial compensation while speaking at events funded by Gilead Sciences, Janssen Cilag, Merck Sharp & Dome and ViiV Healthcare. The other authors declare that they have no competing interests.

Despite the highly effective treatments currently available, hepatitis C virus (HCV) remains one of the major public health concerns related to an infectious agent,

mainly due to a high percentage of individuals remaining undiagnosed (1). Although the incidence of new infections is decreasing with good prevention strategies and curative treatments, models suggest that morbidity will continue to increase over the next few years (2). In addition to these consequences, the burden of disease poses a serious public health challenge to health care systems.

Although the number of newly diagnosed HCV cases in Europe remains high (2), few countries have achieved the World Health Organization (WHO) European Region testing plan target for 2020 (50% of people living with chronic HCV infection being diagnosed with this condition) (1). It is estimated that 3.9 million people are chronically infected with HCV in this territory, with an estimated prevalence of anti-HCV in the general population of 0.1% to 5.9% (3). The actions implemented in HCV screening are partial and limited to some countries. Studies assessing HCV testing practices showed low screening rates (4, 5), which included some influential factors, such as nonreporting by patients of current or historical risk and perceived irrelevance of risk factors in the primary care visit (6).

Screening programs targeting high-risk populations were shown to be cost-effective and more feasible than universal screening, at least in some settings (7). However, given the urgent need to improve the availability and coverage of testing, as well as to address barriers to HCV testing (such as improved policies to include testing guidance), it is advisable to develop screening programs for the general population (1). This study aims to assess which self-reported risk factors are associated with an increased risk of being diagnosed with HCV infection and to create an easy-to-implement scoring system for targeted HCV testing in the general population.

## RESULTS

A total of 7,991 participants were included in the study and filled out the questionnaire; 65.9% were women, and the median age was 43 years (interquartile range [IQR], 33 to 53 years). Most participants (40.6%) had a secondary education, followed by 33.3% with university studies and 25.1% with primary education. The most frequent place of origin was Spain (75.5%), followed by Latin America (15.4%). Of a total of 7,991 participants, 2,894 (36.2%) answered affirmatively to HCV-related questions. A total of 4,717 HCV tests were performed, with 52 (1.1%) being positive. These results have been further described elsewhere (8, 9).

Regarding the population used to create the score, 2,302 patients >50 years of age, 66% male, 94% of Spanish origin, and 6% from Eastern Europe, with a median age of 58 years (IQR, 54 to 62 years), were tested. Of those patients, 867 (38%) had positive questionnaire results and 30 participants had positive HCV test results (1.3% of the tests performed in this population). Of the 30 HCV-positive participants, 22 had HCV RNA detected by PCR. Of those 22, 16 were patients who had been previously diagnosed at some point in the past but had not been linked to care or treatment, and 6 were new diagnoses.

The modeling strategy identified three questions, in addition to gender and place of origin, that showed a C-statistic of 0.896 (95% confidence interval [CI], 0.833 to 0.959). Table 1 shows the model and the scores created from it, while Table 2 shows sensitivity, specificity, and likelihood ratios for the different cutoff points.

Table S2 in the supplemental material shows the distribution of HCV test results for each total score obtained. Based on this, we selected the cutoff point of ≥1, i.e.,

**TABLE 1** Logistic regression model for positive HCV test result

| Parameter | Coefficient (95% CI)$^a$ | P | Score |
|---|---|---|---|
| Male sex | 1.01 (0.18–1.84) | 0.018 | 1 |
| Eastern Europe origin | 1.27 (0.98–2.46) | 0.034 | 1 |
| Have you used intravenous illicit drugs? | 2.93 (1.47–4.38) | <0.001 | 3 |
| Do you think you may have been at risk for acquiring HCV infection through any exposure? | 1.10 (0.26–1.95) | 0.010 | 1 |
| Have you had hepatitis B/C or unexplained liver disease? | 2.98 (2.10–3.86) | <0.001 | 3 |

$^a$Model intercept, −6.43.

**TABLE 2** Sensitivity and specificity with different score cutoff values[a]

| Score cutoff value | Sensitivity (%) | Specificity (%) | LR+ | LR− |
|---|---|---|---|---|
| ≥0 | 100 | 0.0 | 1 | |
| ≥1 | 98.0 | 37.8 | 1.575 | 0.053 |
| ≥2 | 84.0 | 73.2 | 3.139 | 0.219 |
| ≥3 | 76.0 | 89.0 | 6.933 | 0.270 |
| ≥4 | 58.0 | 96.0 | 14.507 | 0.438 |
| ≥5 | 38.0 | 98.7 | 29.845 | 0.628 |
| ≥6 | 26.0 | 99.6 | 58.344 | 0.743 |
| ≥7 | 22.0 | 99.9 | 191.983 | 0.781 |
| ≥8 | 14.0 | 100 | | 0.860 |
| >8 | 0.0 | 100 | | 1 |

[a]LR+, positive likelihood ratio; LR−, negative likelihood ratio.

finding at least one affirmative response, including male gender, Eastern European origin, and three questions in the model. With this cutoff point, we achieved a sensitivity of 98% and a negative likelihood ratio of 0.053, i.e., a 20-fold decrease in the odds of having a positive HCV test result with a negative questionnaire result (score of 0). In our sample of 2,302 participants over the age of 50 years that was used to develop the model, 1,141 (49.8%) had scores of 0 (see Table S2). Using the probability calculator, with a pretest probability of 0.01 (estimated HCV prevalence of 1%), the probability of testing positive for HCV with a score of 0 is 0.00% (95% CI, 0 to 0.001%). If we set a higher prevalence (for example, 5%), then the probability of obtaining a positive test result for HCV with a score of 0 would be 0.00% (95% CI, 0 to 0.02%). We applied this score to the population <50 years of age and obtained a high diagnostic accuracy using the same cutoff point, with estimates akin to those of the model developed for subjects >50 years of age (see Table S3).

## DISCUSSION

This prospective study with a large sample size shows that HCV screening in the general population can be optimized by using a simple 5-item score. By setting the cutoff point at ≥1 for affirmative questions, we achieved high sensitivity and a very low negative likelihood ratio, ruling out HCV infection with a very high probability for participants with scores of 0. In our sample, approximately 50% of participants had scores of 0; therefore, we infer that, with this strategy, one-half of the tests could be dropped by selecting the population with a very low pretest probability of being infected with HCV.

The directed screening for HCV through risk assessment is a strategy to ascertain improved cost-effectiveness of universal screening, accomplished by selecting patients for whom testing is indicated to determine an early diagnosis. This facilitates virological suppression, as well as decreased infection transmission rates. Previous studies identified factors reported by the patient, along with the use of questionnaires. Nguyen et al. found seven factors that were independently associated with HCV infection (10). The Dutch working group, led by Zuure et al., developed a questionnaire for online presentation to establish individuals at risk of HCV infection, as well as offering a free, anonymous test; this achieved sensitivity of <85% and specificity of 64% (11). McGinn et al. validated a questionnaire with 27 items for use in primary care, achieving an area under the receiver operating characteristic (ROC) curve of 0.77, with specificity of 97% in cases with positive responses in three domains (medical history, previous exposure, work history, personal history, or social history) but decreasing to 31% in case of a positive response in only one domain (12). Our shortened questionnaires consist of 5 items, with the advantage of simplicity: any question answered affirmatively indicate that the diagnostic test should be performed. In contrast, we observed a very high sensitivity and a very low negative likelihood ratio for participants who did not exhibit any of the items, so that the questionnaire could easily rule out HCV infection

with high probability. In recent years, the implementation of universal HCV screening has been debated (13). However, most current projects focus on the microelimination of the epidemic in at-risk populations, such as injecting drug users, inmates, or men who have sex with men and engage in high-risk sexual practices (14). WHO defines high-risk populations as those with HCV seroprevalence rates of >5% (15). In 2012, the Centers for Disease Control and Prevention (CDC) in the United States recommended the HCV test for all individuals born between 1945 and 1965, known as Baby Boomers, among whom there is a very high prevalence of HCV infection (16). Since then, several European studies have evaluated the feasibility of targeted screening according to year of birth (17). We recognize that it would be more advisable to perform universal screening to achieve HCV eradication in the cohort of the most prevalent populations. Since this is difficult to strategically achieve, establishing a minimum standard for these criteria is a more attainable goal, with results that would place us in the path of HCV elimination.

This study has some limitations. Risk assessment for HCV was considered exploratory in this study; the development of a subsequent protocol for validation of the questionnaire for HCV infection would be necessary. We found a small number of outcomes (positive anti-HCV results); in addition, the fact that the study was carried out in the general population attending primary care led to a high prevalence of groups with a lower risk of HCV infection, such as women. Nevertheless, this strategy was chosen to prioritize external validity of the score and its implementation to improve screening in the general population. The main strengths of this study are the development of a score that can be easily implemented in daily practice without requiring the use of calculators or consuming much of the health care provider's time. In addition, the score has shown good performance in a cohort of patients who were not used in its development, reinforcing its validity for use in the general population.

**Conclusion.** Targeted HCV screening, through a simple risk assessment questionnaire, can be a useful strategy and can be seen as a cost-effective alternative to universal testing. With this tool, we achieved high diagnostic accuracy; this can be implemented in clinical practice to guide HCV test requests with a very low probability of excluding infected individuals, optimizing resources when it is not feasible to perform universal screening.

## MATERIALS AND METHODS

This is a substudy derived from a prospective, randomized 1:1, clustered, crossover study carried out in four primary care centers of the basic health area of the Ramón y Cajal Hospital in Madrid (Spain), between November 2016 and September 2017. The study statistician randomized the centers to the intervention (questionnaire and rapid tests) or to the control arm (usual care). A cluster randomization strategy was used to facilitate fieldwork and to avoid contamination and possible errors due to the large number of professionals and participants to be included. The comparison of these centers in terms of geographic characteristics, population, and providers is shown in Table S1 in the supplemental material. The initial work evaluated combined HCV and HIV screening and was previously published (8). Here, we present the specific results regarding the development of a questionnaire for targeted HCV screening. The study was approved by the Research Ethics Committee at the Ramón y Cajal Hospital and the Madrid Regional Primary Care Research Committee, with all participants giving written informed consent.

We prospectively included participants between 18 and 70 years of age who were not infected with HIV and who had attended one of the primary care centers for any reason. Participation was offered to all persons who met the inclusion criteria and who attended any of the four health centers within the period determined for each of them until the predetermined sample size was completed. We adapted a previously validated questionnaire to assess the risk of HIV infection (18) to then incorporate HCV screening after reaching a consensus with a group of experts in HIV/HCV coinfection. The questionnaire consists of 6 items that investigate the risk of exposure to HIV or HCV (based on identified routes of transmission) and 15 items related to HIV indicator conditions (based on a selection of the HIV Indicator Diseases across Europe Study [HIDES]) (19), of which 6 were selected for HCV (see the supplemental material). The questionnaire was self-completed by the participants. However, five specialized nurses were part of the research team and helped complete the questionnaire if there were any doubts. After completion of this 21-item questionnaire, point-of-care HCV screening was performed using rapid tests (anti-HCV WB/S/P test; TürkLab Laboratories, Izmir, Turkey) for subjects with at least one positive response to HCV-related questions and for all subjects >50 years of age, even if they did not answer positively, since

this age group corresponds to the cohort with the greatest known prevalence of HCV infection in our setting.

We created a logistic regression model to assess the risk of having a positive HCV test result. For this purpose, the population >50 years of age was used, since those subjects were participants for whom the test was universally performed, regardless of questionnaire results. Candidate predictors, including sex and place of origin, were selected according to knowledge from previous literature. The covariates included in the model were selected by backward stepwise selection, keeping in the model variables with $P$ values of <0.05. Discrimination of models was performed using the C-statistic following the parsimony criterion, which selects the model with the smallest number of variables. We then calculated the score for each variable using the rounded value of the quotient resulting from dividing the coefficient of each category by the lowest coefficient. We calculated the best cutoff point to maintain high sensitivity, targeting >90%, as well as a low negative likelihood ratio, targeting <0.05. Predictive values are of limited value in this case, because they depend on the prevalence of the disease. We then performed a sensitivity analysis by applying the score obtained to the population <50 years of age, to assess its diagnostic accuracy. We used a diagnostic studies calculator (20) for simulations with different pretest probabilities, according to HCV prevalence in other areas. All statistical analyses were performed using Stata v. 17.0 (StataCorp LP, College Station, TX, USA).

## SUPPLEMENTAL MATERIAL

Supplemental material is available online only.

**SUPPLEMENTAL FILE 1**, PDF file, 0.1 MB.

## ACKNOWLEDGMENTS

This study was supported by three competitive grants, i.e., Instituto de Salud Carlos III (Plan Estatal de I+D+i2013–2016) grants PI12-00995 and PI16-00551 and Ministerio de Sanidad, Servicios Sociales e Igualdad project EC11-144, all cofinanced by the European Development Regional Fund (A way to achieve Europe) (ERDF), partially funded by the RD16/0025/0001 project as part of the Plan Nacional R+D+I and cofinanced by ISCIII-Subdirección General de Evaluación y Fondo Europeo de Desarrollo Regional (FEDER).

J.M.-S. reports personal fees from ViiV Healthcare, Janssen Cilag, Gilead Sciences, and Merck Sharp & Dohme and nonfinancial support from ViiV Healthcare, Jannsen Cilag, and Gilead Sciences, outside the submitted work. B.R. has received funding to attend conferences from Gilead and obtained a funded project from Roche. J.C.G. has been involved in speaking activities from Gilead Sciences, GeneExpert, and Hologic. He received grants from Merck Sharp & Dohme, Abbvie, Abbott Molecular, and Roche. He has done consulting work for Abbott Molecular and Gilead Sciences. S.M. has been involved in speaking activities and received grants for research from Abbott, Boehringer Ingelheim, Bristol-Myers Squibb, Gilead Sciences, GlaxoSmithKline, Janssen Cilag, Merck Sharp & Dohme, Pfizer, Roche, and Schering-Plough. M.J.P.-E. has done consulting work for Abbvie, Boehringer Ingelheim, ViiV Healthcare, Gilead Sciences, and Janssen Cilag; she has received fellowships for clinical research from ViiV Healthcare, Gilead Sciences, Janssen, and Merck Sharp & Dome and financial compensation while speaking at events funded by Gilead Sciences, Janssen Cilag, Merck Sharp & Dome, and ViiV Healthcare. The other authors declare that they have no competing interests.

The DRIVE 03 Study Group includes C. Gómez-Ayerbe, M. Sánchez Conde, E. Loza, S. del Campo, A. Sánchez, A. Moreno, and M. Rodríguez (Hospital Universitario Ramón y Cajal, Madrid, Spain); A. Cano, A. Fernández, M. E. Calonge, and C. Santos (C. S. García Noblejas, Madrid, Spain); S. Ares (C. S. Mar Báltico, Madrid, Spain); C. Labrador (Hospital de la Princesa, Madrid, Spain); P. González (SUMMA, Madrid, Spain); L. Polo (Hospital General Universitario Gregorio Marañón, Madrid, Spain); and Y. de la Fuente (C. S. Aquitania, Madrid, Spain).

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
