## [Reviewer comments · Microbiology Spectrum]

Microbiology Spectrum

An easy-to-implement risk score for targeted hepatitis C testing in the general population

Javier Martínez-Sanz, María Jesús Vivancos-Gallego, Borja Manuel Fernández-Félix, Alfonso Muriel, Pilar Pérez-Elías, Almudena Uranga, Beatriz Romero-Hernandez, Juan Carlos Galán, Santiago Moreno, and María Jesús Pérez-Elías

Corresponding Author(s): Javier Martínez-Sanz, Hospital Universitario Ramón y Cajal

Review Timeline:

Submission Date:	November 17, 2021
Editorial Decision:	January 23, 2022
Revision Received:	January 24, 2022
Accepted:	February 19, 2022

Editor: Francisco Uzal

Reviewer(s): Disclosure of reviewer identity is with reference to reviewer comments included in decision letter(s). The following individuals involved in review of your submission have agreed to reveal their identity: Karen Scott (Reviewer #1)

Transaction Report:

DOI: <https://doi.org/10.1128/spectrum.02286-21>

January 23, 2022

Dr. Javier Martínez-Sanz
Hospital Universitario Ramón y Cajal
Infectious Diseases
M-607, km. 9,100
Madrid 28034
Spain

Re: Spectrum02286-21 (An easy-to-implement risk score for targeted hepatitis C testing in the general population)

Dear Dr. Javier Martínez-Sanz:

Link Not Available

Sincerely,

Francisco Uzal

Journals Department
Reviewer comments:

Reviewer #1 (Comments for the Author):

Thank you for inviting me to review this paper on a developing a risk score for targeted hepatitis C testing in the general population. The subject matter was interesting and the focus of the study addresses an important question. The study and results are well described and would be improved with some greater differentiation between the original study and this sub-study.

I have a few major and minor comments, detailed below.

Major comments:

- All respondents aged over 50 years were antibody tested and data from over 50s were included in the logistic regression. It would be informative to add the justification for why this cut off was selected.
- In the methods it is stated that this is a prospective, randomised 1:1, clustered, crossover study, carried out in four primary care centers. Nothing further is said about how the randomisation was carried out. No further details about the clusters or crossover are provided. This design relates to the original study (described in Martínez-Sanz, Javier et al. "Hepatitis C and HIV combined screening in primary care: A cluster randomized trial." Journal of viral hepatitis vol. 28,2 (2021): 345-352.

doi:10.1111/jvh.13413) rather than the work done for this sub-study. In the original study the intervention arm included a 4-hour educational programme, the use of a risk-assessment questionnaire and rapid tests. In the control centres, only the educational intervention was provided. That the survey respondents had 4 hours of HCV and HIV education before completing the survey should be mentioned in this paper as this may influence the way the questionnaire is completed and this would not happen in the usual primary care setting.

- It is important to describe how people attending the clinic were selected to take part in the study. For example, was everyone who was eligible asked to take part during a specified time frame?
- It would be informative to have some information about how the questionnaires were completed. Were participants interviewed or was the questionnaire self-completed?
- The question "Do you think you may have been at risk for acquiring HCV infection through any exposure?" does not appear on the questionnaire for HCV risk assessment provided in the supplementary materials. There is a similar question about HIV but not one about HCV, perhaps the wrong questionnaire has been included?
- I am not a statistician but I found the statistical methods to be well described and appropriate. I have a concern over the low number of outcomes (30 antibody positive test results) which could mean the analysis is underpowered. This could be mentioned as a limitation.
- It would be interesting to know the proportion of respondents with positive antibody results who also had a positive PCR.

Minor points:

- The statement that "A total of 2,302 questionnaires were used." on line 39 of the abstract could be clarified. The statement is misleading and I think what is meant that data collected from 2302 participants were included in the analysis.
- Abbreviations have been used in the abstract without being written out in full on first use.

Reviewer #2 (Comments for the Author):

This paper evaluated an easy-to-implement risk score for targeted hepatitis C (HCV) testing in the general population. More than 2000 persons completed a 21-item questionnaire, followed by rapid anti-HCV testing. The topic of research is pertinent and deserves further development of cost-effective strategies to eliminate HCV. The major limitation is that the strategy was not validated.

Specific comments:

1. Introduction, line 72: Consider whether it should say "an increased diagnosis of HCV oinfection" where it says "an increased risk of HCV infection".
2. Methods: The questionnaire was adapted from a previously validated one reported in reference 9, but that study refers to HIV risk. having reached a consensus with a group of experts in HIV/HCV is vague, there are no references or further explanation, and in any case the co-infection HIV/HCV is not suitable to address the score to the general population as it was done.
3. Methods: There is no justification or calculation of the sample size.
4. Methods: The justification to consider the population over 50 years "since they were participants on whom the test was universally performed regardless of questionnaire results" is odd and does not fit with the general design of the study. If only those over 50 were considered, why participants between 18 and 50 years were included? If those over 50 had the HCV test universally performed, why then the assessment of this strategy (which indeed was not followed in this age segment!)?
5. Results: Nearly 8000 participants contributed with questionnaires, but only 2302 were used to create the score. In addition, these 2302 were aged over 50 years. Although the justification to consider the population over this age is explained in the methods, this information is not contained within the title or the abstract.

Staff Comments:

Preparing Revision Guidelines

Please return the manuscript within 60 days; if you cannot complete the modification within this time period, please contact me. If you do not wish to modify the manuscript and prefer to submit it to another journal, please notify me of your decision immediately so that the manuscript may be formally withdrawn from consideration by Microbiology Spectrum.

RESPONSE TO REVIEWERS

Manuscript ID: Spectrum02286-21

Article title: *An easy-to-implement risk score for targeted hepatitis C testing in the general population*

The authors would like to thank the reviewers and editors for their careful review of our manuscript and their constructive comments and suggestions for improving its quality. The following responses have been prepared to address all of the referees' comments point-by-point. Line numbers refer to the revised manuscript with tracked changes.

REVIEWERS' COMMENTS FOR THE AUTHOR:

Reviewer #1

Thank you for inviting me to review this paper on a developing a risk score for targeted hepatitis C testing in the general population. The subject matter was interesting and the focus of the study addresses an important question. The study and results are well described and would be improved with some greater differentiation between the original study and this sub-study.

Authors: We thank the Reviewer for the positive assessment of our work.

I have a few major and minor comments, detailed below.

Major comments:

1. All respondents aged over 50 years were antibody tested and data from over 50s were included in the logistic regression. It would be informative to add the justification for why this cut off was selected.

Authors: The present work corresponds to a sub-study carried out on the basis of the previously published work referenced in the manuscript (Reference 8), which for reasons of word count has not been detailed in the present work. However, thanks to the Reviewer's concerns, we have become aware of the need to further explain the initial work in order to clarify the scenario on which the present work is based.

In the initial work (summarized on lines 77-113), HCV testing was performed on all individuals over 50 years of age, regardless of the questionnaire result. This age group corresponds to the cohort with the highest known prevalence of HCV infection in our setting, and we set out as a secondary objective to explore whether there were any stated risk factors, in addition to age, among participants diagnosed with HCV infection. Taking advantage of the fact that we have this cohort of patients in whom the questionnaire and also the rapid tests were performed, we created the model presented in this work. Subsequently, as explained in the manuscript (line 132-133) we performed a sensitivity analysis by applying the obtained score to the population under 50 years to assess its performance, obtaining a high diagnostic accuracy using the same cut-off point, with estimates akin to those of the model developed for those over 50 years of age. Following the Reviewer's recommendation, this information has been clarified in the manuscript (lines 112-117).

2. In the methods it is stated that this is a prospective, randomised 1:1, clustered, crossover study, carried out in four primary care centers. Nothing further is said about how the randomisation was carried out. No further details about the clusters or crossover are provided. This design relates to the original study (described in Martínez-Sanz, Javier et al. "Hepatitis C and HIV combined screening in primary care: A cluster randomized trial." Journal of viral hepatitis vol. 28,2 (2021): 345-352. doi:10.1111/jvh.13413) rather than the work done for this sub-study. In the original study the intervention arm included a 4-hour educational programme, the use of a risk-assessment questionnaire and rapid tests. In the control centres, only the educational intervention was provided. That the survey respondents had 4 hours of HCV and HIV education before completing the survey should be mentioned in this paper as this may influence the way the questionnaire is completed and this would not happen in the usual primary care setting.

Authors: As indicated in response #1, the information on the original paper from which this cohort of patients was obtained has been clarified and expanded, including information on randomization and clustering (lines 80-83). To avoid confusion, it has been remarked that the original work has been previously published and the complete methods can be consulted. The methods corresponding to this work have been clarified (lines 77-113).

3. It is important to describe how people attending the clinic were selected to take part in the study. For example, was everyone who was eligible asked to take part during a specified time frame?

Authors: As indicated by the Reviewer, participation was offered to all persons who met the inclusion criteria and who attended any of the four health centers within the period determined in each of them, until the predetermined sample size was completed (information available in Table S1 of the Supplementary Material). This information has been added in the Methods section (lines 92-95).

4. It would be informative to have some information about how the questionnaires were completed. Were participants interviewed or was the questionnaire self-completed?

Authors: The questionnaire was self-completed by the participants. However, five specialized nurses were part of the research team and helped to complete the questionnaire in case there were any doubts. These same nurses were in charge of performing the rapid tests afterward. This information has been added to the manuscript (lines 106-108).

5. The question "Do you think you may have been at risk for acquiring HCV infection through any exposure?" does not appear on the questionnaire for HCV risk assessment provided in the supplementary materials. There is a similar question about HIV but not one about HCV, perhaps the wrong questionnaire has been included?

Authors: Thank you for highlighting this error, carried over from previous work. The correct question is "Do you think you may have been at risk of acquiring HIV or HCV infection through any exposure?". It has been corrected in the Supplementary Material.

6. I am not a statistician but I found the statistical methods to be well described and appropriate. I have a concern over the low number of outcomes (30 antibody positive test results) which could mean the analysis is underpowered. This could be mentioned as a limitation.

Authors: Following the reviewer's recommendations, we have added to the limitations the low prevalence of positive tests (lines 225-226). However, please note that, as opposed to predictive values, sensitivity and likelihood ratios are not affected by disease prevalence and therefore these results could be adopted to other patient populations, regardless of whether the prevalence is different.

7. It would be interesting to know the proportion of respondents with positive antibody results who also had a positive PCR.

Authors: Of the total 30 anti-HCV-positive participants, 22 had HCV RNA detection by PCR. Of these 22, 16 were patients who had been previously diagnosed at some point in the past but had not been linked to care or treated, and 6 were new diagnoses. All were treated with direct-acting antiviral agents achieving sustained viral response. This information has been added to the manuscript (lines 151-157)

Minor points:

8. The statement that "A total of 2,302 questionnaires were used." on line 39 of the abstract could be clarified. The statement is misleading and I think what is meant that data collected from 2302 participants were included in the analysis.

Authors: The statement has been changed to "Data collected from 2302 participants were included in the analysis"

9. Abbreviations have been used in the abstract without being written out in full on first use.

Authors: We have reviewed and written out the first in-text reference to all acronyms.

Reviewer #2:

This paper evaluated an easy-to-implement risk score for targeted hepatitis C (HCV) testing in the general population. More than 2000 persons completed a 21-item questionnaire, followed by rapid anti-HCV testing. The topic of research is pertinent and deserves further development of cost-effective strategies to eliminate HCV. The major limitation is that the strategy was not validated.

Authors: We thank the reviewer for the positive assessment of the relevance of our work. Indeed, so far, the strategy has not been validated in a different cohort, however, it has been applied to the population of participants under 50 years of age, which was not used for the development of the score and can therefore be understood as a validation group, obtaining similar results (Supplementary Table S3).

Specific comments:

1. Introduction, line 72: Consider whether it should say "an increased diagnosis of HCV coinfection" where it says "an increased risk of HCV infection".

Authors: To avoid confusion, in accordance with the Reviewer's recommendations, we have changed the wording to "an increased risk of being diagnosed with HCV infection" (line 73)

2. Methods: The questionnaire was adapted from a previously validated one reported in reference 9, but that study refers to HIV risk. Having reached a consensus with a group of experts in HIV/HCV is vague, there are no references or further explanation, and in any case the co-infection HIV/HCV is not suitable to address the score to the general population as it was done.

Authors: As the Reviewer comments, the 21-item questionnaire was previously validated for HIV infection. In order to take advantage of the same questionnaire to assess the risk of HCV infection, some questions were selected according to previous knowledge on the subject. The current score can therefore only be applied to HCV infection and not to HIV/HCV coinfection (for HIV the full 21-item questionnaire should be applied). Regarding the validation of the results, as mentioned before, the population under 50 years of age has been used as a validation cohort, but the application of this score in different populations to see its performance remains for a future work. We have clarified the Methods section (lines 77-113) to avoid possible misinterpretation.

3. Methods: There is no justification or calculation of the sample size.

Authors: For a power of 80% and an alpha error of 0.05, a total of 6883 participants were estimated (Stata code: `nsizen co2p, p0(0.15) p1(0.45) beta(20)`)

*Proportion (%) of events in: Group 0 = .15%
Group 1 = .4%
Minimum expected effect size: Difference= .25%
Alpha Risk = 5%; Beta Risk = 20% (Power = 80%)*

METHOD	SAMPLE SIZE by group	
	Two-Sided Test	One-Sided Test
Normal	6887	5425
ArcoSinus	6513	5130

However, this sample size applies to the original previously published work from which this cohort of participants is drawn (reference #8). In this study, the overall number of participants with a completed questionnaire and a performed test was used, so no specific sample size was calculated.

4. Methods: The justification to consider the population over 50 years "since they were participants on whom the test was universally performed regardless of questionnaire results" is odd and does not fit with the general design of the study. If only those over 50 were considered, why participants between 18 and 50 years were included? If those over 50 had the HCV test universally performed,

why then the assessment of this strategy (which indeed was not followed in this age segment!)?

Authors: Please see response #1 to Reviewer #1. The present work corresponds to a substudy carried out on the basis of the previously published work referenced in the manuscript (Reference 8), which for reasons of word count has not been detailed in the present work. However, thanks to the Reviewers' concerns, we have become aware of the need to further explain the initial work in order to clarify the scenario on which the present work is based. In the initial work (summarized on lines 77-113), HCV testing was performed on all individuals over 50 years of age, regardless of the questionnaire result. This age group corresponds to the cohort with the highest known prevalence of HCV infection in our setting, and we set out as a secondary objective to explore whether there were any stated risk factors, in addition to age, among participants diagnosed with HCV infection. Taking advantage of the fact that we have this cohort of patients in whom the questionnaire and also the rapid tests were performed, we created the model presented in this work. Subsequently, as explained in the manuscript (line 132-133) we performed a sensitivity analysis by applying the obtained score to the population under 50 years to assess its performance, obtaining a high diagnostic accuracy using the same cut-off point, with estimates akin to those of the model developed for those over 50 years of age. Following the Reviewers' recommendation, this information has been clarified in the manuscript (lines 112-117, 132-133).

5. Results: Nearly 8000 participants contributed with questionnaires, but only 2302 were used to create the score. In addition, these 2302 were aged over 50 years. Although the justification to consider the population over this age is explained in the methods, this information is not contained within the title or the abstract.

Authors: Please see previous response #4. Following the Reviewer's recommendations, this information has been added to the abstract. The abstract has been expanded with additional information, and the "Importance" section has been created to provide a non-technical explanation of the significance of the study to the field.

February 19, 2022

Dr. Javier Martínez-Sanz
Hospital Universitario Ramón y Cajal
Infectious Diseases
M-607, km. 9,100
Madrid 28034
Spain

Re: Spectrum02286-21R1 (An easy-to-implement risk score for targeted hepatitis C testing in the general population)

Dear Dr. Javier Martínez-Sanz:

Your manuscript has been accepted, and I am forwarding it to the ASM Journals Department for publication. You will be notified when your proofs are ready to be viewed.

Sincerely,

Francisco Uzal
Editor, Microbiology Spectrum
